# Wasp Venom Biochemical Components and Their Potential in Biological Applications and Nanotechnological Interventions

**DOI:** 10.3390/toxins13030206

**Published:** 2021-03-12

**Authors:** Aida Abd El-Wahed, Nermeen Yosri, Hanem H. Sakr, Ming Du, Ahmed F. M. Algethami, Chao Zhao, Ahmed H. Abdelazeem, Haroon Elrasheid Tahir, Saad H. D. Masry, Mohamed M. Abdel-Daim, Syed Ghulam Musharraf, Islam El-Garawani, Guoyin Kai, Yahya Al Naggar, Shaden A. M. Khalifa, Hesham R. El-Seedi

**Affiliations:** 1 Agricultural Research Centre, Department of Bee Research, Plant Protection Research Institute, Giza 12627, Egypt; aidaabd.elwahed@arc.sci.eg; 2Department of Chemistry, Faculty of Science, Menoufia University, Shebin El-Kom 32512, Egypt; nermeen.yosri@science.menofia.edu.eg; 3School of Food and Biological Engineering, Jiangsu University, Zhenjiang 212013, China; haroona28@yahoo.com; 4Department of Zoology, Faculty of Science, Menoufia University, Shebin El-Kom 32512, Egypt; hanem.sakr@science.menofia.edu.eg (H.H.S.); dr.garawani@science.menofia.edu.eg (I.E.-G.); 5National Engineering Research Center of Seafood, School of Food Science and Technology, Dalian Polytechnic University, Dalian 116034, China; duming@dlpu.edu.cn; 6Alnahalaljwal Foundation Saudi Arabia, P.O. Box 617, Al Jumum 21926, Makkah, Saudi Arabia; ahmed@alnahalaljwal.com.sa; 7College of Food Science, Fujian Agriculture and Forestry University, Fuzhou 350002, China; zhchao@live.cn; 8State Key Laboratory of Quality Control in Chinese Medicine, Institute of Chinese Medical Sciences, University of Macau, Taipa, Macau, China; 9Department of Medicinal Chemistry, Faculty of Pharmacy, Beni-Suef University, Beni-Suef 62514, Egypt; ahmed.abdelazeem@pharm.bsu.edu.eg; 10Department of Pharmaceutical Sciences, College of Pharmacy, Riyadh Elm University, Riyadh 11681, Saudi Arabia; 11Abu Dhabi Food Control Authority, Al Ain 52150, United Arab Emirates; saad.masry@adafsa.gov.ae; 12Department of Plant Protection and Biomolecular Diagnosis, Arid Lands Cultivation Research Institute (ALCRI), City of Scientific Research and Technological Applications, New Borg El-Arab City, Alexandria 21934, Egypt; 13Pharmacology Department, Faculty of Veterinary Medicine, Suez Canal University, Ismailia 41522, Egypt; abdeldaim.m@vet.suez.edu.eg; 14International Center for Chemical and Biological Sciences, H.E.J. Research Institute of Chemistry, University of Karachi, Karachi 75270, Pakistan; musharraf@iccs.edu; 15Laboratory of Medicinal Plant Biotechnology, College of Pharmacy, Zhejiang Chinese Medical University, Hangzhou 310053, China; kaiguoyin@163.com; 16General Zoology, Institute for Biology, Martin Luther University Halle-Wittenberg, Hoher Weg 8, 06120 Halle (Saale), Germany; yehia.elnagar@science.tanta.edu.eg; 17Zoology Department, Faculty of Science, Tanta University, Tanta 31527, Egypt; 18Department of Molecular Biosciences, Stockholm University, the Wenner-Gren Institute, SE-106 91 Stockholm, Sweden; 19International Research Center for Food Nutrition and Safety, Jiangsu University, Zhenjiang 212013, China; 20Division of Pharmacognosy, Department of Pharmaceutical Biosciences, Uppsala University, Biomedical Centre, P.O. Box 574, 751 23 Uppsala, Sweden

**Keywords:** wasp’s venom, biomedical properties, bioactive compounds, nanotechnology applications, allergy

## Abstract

Wasps, members of the order Hymenoptera, are distributed in different parts of the world, including Brazil, Thailand, Japan, Korea, and Argentina. The lifestyles of the wasps are solitary and social. Social wasps use venom as a defensive measure to protect their colonies, whereas solitary wasps use their venom to capture prey. Chemically, wasp venom possesses a wide variety of enzymes, proteins, peptides, volatile compounds, and bioactive constituents, which include phospholipase A2, antigen 5, mastoparan, and decoralin. The bioactive constituents have anticancer, antimicrobial, and anti-inflammatory effects. However, the limited quantities of wasp venom and the scarcity of advanced strategies for the synthesis of wasp venom’s bioactive compounds remain a challenge facing the effective usage of wasp venom. Solid-phase peptide synthesis is currently used to prepare wasp venom peptides and their analogs such as mastoparan, anoplin, decoralin, polybia-CP, and polydim-I. The goal of the current review is to highlight the medicinal value of the wasp venom compounds, as well as limitations and possibilities. Wasp venom could be a potential and novel natural source to develop innovative pharmaceuticals and new agents for drug discovery.

## 1. Introduction

Vespid wasps (Family: Vespidae) are distributed worldwide and comprise more than 5000 species. Wasp venom has a wide variety of chemical constituents, which includes proteins, peptides (e.g., mastoparan, eumenitin, eumenitin-R, rumenitin-F, EpVP, decoralin, and anoplin), enzymes (hyaluronidase, α-glucosidase, phosphatase phospholipase A2, and phospholipase B), and small molecules [1,2,3]. The isolated compounds from wasp venom have shown several beneficial activities such as antimicrobial [4,5], anticancer [6], and anti-inflammatory effects [7]. However, their peptides have been presented in trace quantities. Solid phase peptides synthesis (SPPS) was attributed to the design and development of these molecules [8]. Successfully, several peptides and their analogues were synthesized via SPPS technology such as mastoparan [9], anoplin [10], decoralin [11], polybia-MP-I [12], polybia-CP [13,14], polydim-I [15], and agelaia-MP [16]. The synthetic peptides have antimicrobal, and anticancer properties [17,18].

The nests and venoms of wasps have been their role in the synthesis of nanoparticles of gold and silver tested. These nanoparticles were proven effective as antimicrobial and anticancer entities against a variety of microorganisms and cancer cells [19,20,21].

Despite preliminary medicinal outcomes, the interaction between wasp venom and human organs is still under debate. Wasp venom impacts the physiological aspects of the human body and could also lead to an allergic reaction [22].

Allergic reaction to wasp venom is a devastating problem due to the progressing immune responses of different systems. For instance, *Vespa velutina* venom administration lead to the failure of multi-organisms and even death among the Chinese population; and that was mostly due to toxins that are usually known to cause pain, inflammation, kidney and liver failure, cardiac arrhythmia, and sometimes neurotoxicity. Thus, many efforts are being invested into combating the allergic reactions and improving life quality using venom immunotherapy (VIT) [23]. VIT is the most effective method known so far for the avoidance of the systemic sting reactions even after discontinuation of the therapy [24].

This review attempts to shed light on the biochemical properties and potential applications of wasp venom. Previous studies have suggested that venom peptides could contribute to biological activities, such as anticancer, antimicrobial, neuroprotective, anti-inflammatory, and antioxidant effects. It is important to investigate the components of venom to learn how the underlying defensive mechanisms have evolved since evidence suggests that the wasp venom can differ internally and interspecifically from other types of venom. Isolation and characterization of the venom components are essential steps in understanding the envenoming process [25].

## 2. Biological Properties of Wasp Venom, and Their Isolated and Synthesized Bioactive Peptides

### 2.1. Biological Properties

Studies have been conducted on venomous wasp structures, and their mode of action dating back to over 50 years ago. However, the therapeutic value of these toxins remains relatively unexplored. Further experiments are needed to fill the gap, and implementat quality control to elucidate wasp venom biological properties. As shown below, wasp venom exhibits biological properties, including antimicrobial, anticoagulant, genotoxic, and anti-inflammatory properties (Figure 1) [26,27,28,29].

#### 2.1.1. Antimicrobial Activities

Today, microbial infections are a significant human concern globally. The emergence of infectious diseases and the scarcity of vaccines pose a significant danger to human health; thus, there is an immediate need to develop new antimicrobial agents [30]. *Vespa orientalis*’s crude venom contains peptides and proteins. The venom has antimicrobial activity against Gram-positive and Gram-negative bacteria at very low concentrations relative to tetracycline (positive control). The inhibition zones were 10.2, 12.6, 22.4, and 22.7 mm for *Klebsiella pneumonia*, *Staphylococcus aureus, Escherichia coli*, and *Bacillus subtilis*, respectively, while MIC values were 128, 64, 64, and 8 μg/mL, respectively. The MIC_50_ and MIC_90_ values were 74.4 and 119.2 μg/mL for *K. pneumonia*, 63.6 and 107 μg/mL for *S. aureus*, 45.3 and 65.7 μg/mL for *E. coli*, and 4.3 and 7.0 μg/mL for *B. subtilis*, respectively [31]. Previous studies have determined that the venom from Parischnogaster, Liostenogaster, Eustenogaster, and Metischnogaster wasps inhibited the development of Gram-positive *B. subtilis*, Gram-negative *E. coli*, and *Saccharomyces cerevisiae* yeast [32]. The peptide mastoparan-c, derived from *Vespa crabro* venom, triggered antimicrobial action toward resistant strains of *S. aureus* (Gram-positive) bacteria [26].

#### 2.1.2. Anti-Inflammatory Activities

Inflammation is an underlying cause of several destructive disorders such as arthritis, cancer, and asthma. Anti-inflammatory medications are currently used to suppress short- and long-term body responses, and thus, it is vital to recognize new molecules with similar properties [33]. *Vespa tropica* venom effectively reduced oxidative stress and stimulated microglia via lipopolysaccharides (LPS) release. Wasp venom treatment (5 and 10 μg/mL) greatly attenuated LPS induced activation of NF-kB phosphorylation [34]. *Bracon hebetor* venom (BHV) affected LPS-induced nitric oxide (NO) in RAW 264.7 cells and septic shock in mouse models. BHV strongly mediated LPS-induced inflammation without any cytotoxicity at a concentration of 0.1–0.4 μg/mL [35]. Moreover, *Nasonia vitripennis* venom contains at least 80 proteins, and it exerts anti-inflammatory impacts via down-regulation of the proinflammatory cytokine IL-1β [27].

#### 2.1.3. Genotoxicity

*Polybia paulista* wasp venom concentrations below 0.01–10 μg/mL did not cause cytotoxicity and showed genotoxic and mutagenic potential in HepG2 cells. The genotoxic and mutagenic behavior of *P. paulista* venom could be explained by the action of phospholipase, mastoparan, and hyaluronidase, leading to cell membrane disruption and genetic material alterations or even DNA mutations [29].

#### 2.1.4. Anticoagulant

The venom of *Polybia occidentalis*, a social wasp, has anticoagulant, and fibrinogen-degrading pharmacological properties. Anticoagulation occurs at different stages of the clotting process (intrinsic, extrinsic, and specific pathway). Venom can inhibit platelet aggregation and destroy plasma fibrinogens [28].

### 2.2. Isolated and Synthesized Bioactive Peptides from Wasp Venoms

Wasp venoms are cocktails of peptides, proteins, and small organic molecules like volatiles compounds (Figure 1 and Figure 2), where peptides are the most abundant compounds, as mentioned in Table 1 [36,37]. The minute quantity of extracted venom stands as a hindrance to the analysis and understanding of the pharmacological, biological, and ecological aspects of the venom constituents. Here, we discuss the isolated peptides from wasp venom and their chemical design via SPPS [8].

#### 2.2.1. Mastoparans

The mastoparans are comprised of a class of peptides isolated from *Vespula lewisii* [38], *V. crabro* [26], *Vespula vulgaris* [4], and *Polistes jadwigae* [39]. Mastoparans are characterized by their antitumor activity against melanoma cells (B16F10-Nex2) [38].

##### Mastoparan (MP)

Mastoparan (MP), a major component of *P. jadwigae* wasp venom, is a basic amphiphilic *α*-helical peptide that consists of 14 amino acid residues, hydrophobic and essential amino acids, and an amino acid *C*-terminus, as shown in Table 1 [39]. These characters are specific for the cationic amphiphilic peptide (CAP) class and favour the α-helix conformation while in contact with bilayer phospholipids [40]. MP has several biological effects and has shown antimicrobial properties [41], increased histamine release from mast cells [42], and cytotoxicity effect on tumor cells [18]. MP-induced mitochondrial permeability and powerful transition of mitochondrial permeability (PT) in a range of 25 μM in a homogeneous K562 cell are reported [43]. Moreover, MP exerts anticancer activities toward leukemia, myeloma, and breast cancer cells. In a mouse model of mammary carcinoma, MP and gemcitabine (drug) worked synergistically [18]. MP was active on a dose-dependent basis with doses ranging from 77.9 to 432.5 μM against human cancer cells (A2058 (melanoma), SiHa (cervical carcinoma), Jurkat (T cell leukemia), MCF-7, MDA-MB-231, and SK-BR-3 (breast cancer). The IC_50_ of B16F10 murine melanoma was 165 μM. MP-induced apoptosis involves activation of caspase −9, −12, and −3, PARP cleavage, upregulation of pro-apoptotic Bax, and Bim, down-regulation of anti-apoptotic Bcl-XL; furthermore, cell apoptosis induced mitochondrial membrane disruption [38].

MP inhibited bradykinin-induced phosphoinositide hydrolysis within 5 min of administration at a concentration of 30 μM and induced the release of prostaglandin E2 (PGE 2) in rabbit astrocytes within 10 min [44].

The synthetic peptide derived from MP is called mastoparan ([I5, R8] MP) and has a wide range of antimicrobial activities against bacteria and fungi at MIC values of 3–25 µM with no hemolytic or cytotoxic properties to the human embryonic kidney cell line (HEK-293 cells). The synthesis does not appear to change the α-helical conformation but enhances the biological activity [45]. Ten MP derivatives have been synthesized via SPPS strategies and evolved against *Acinetobacter baumannii*. MP analogs (H-INIKALAALAKKII-NH_2_, H-INLKALAALAKKIL-CH_2_CH_2_NH_2,_ and Gu-INLKALAALAKKIL-NH_2_) demonstrated the same behavior against *A. baumannii* as the original peptide (2.7 µM) and retained its consistency in the presence of human serum for more than 24 h [9]. Three MP analogs, MK4589 (INWKKIAKKVAGML-NH_2_), MK45789 (INWKKIKKKVAGM), and MK4578911 (INWKKIKKKVKGML-NH_2_), were synthesized, and exhibited strong antibacterial properties against Gram-negative bacteria compared to the reference antibiotic, chloramphenicol [46]. Mastoparan-V1 (MP-V1), *a de novo* type of *V. vulgaris* venom mastoparan, has higher anti-Salmonella activity than other mastoparans [4]. MP analog peptides showed activity against *Candida albicans*, with low cytotoxicity and non-teratogenicity using cell cultures and zebrafish models [47]. Synthetic MP-V1 has antimicrobial properties at MICsvalues of 106.95, 56.86, and 123 µg/mL against *Salmonella Gallinarum*, *S. typhimurium*, *and S. enteritidis*, respectively [48].

##### Mastoparan-B (MP-B)

Mastoparan-B (MP-B), the mastoparan homolog of *Vespa basalis* venom, has a less hydrophobic amino acid sequence with four lysines (LKLKSIVSWAKKVL-NH_2_) [49] and is approved as a cardiovascular depressor [50] and antibacterial agent [51]. MP-B shows powerful hemolytic activity secondary to the stimulation of histamine release from rat peritoneal mast cells [49]. A synthetic MP-B analog (LDLKSIVSWAKKVL-NH_2_), in which lysine was replaced by asparagine at position 2, showed a remarkable decline of cardiovascular depressors; in contrast, the analog with leucine replacing lysine at position 4, 11, or 12 (LKLLSIVSWALLVL-NH_2_) did not display the same effect [50].

##### Mastoparan-M

Mastoparan-M is an amphipathic tetradecapeptide toxin and a vespid venom mastoparan counterpart isolated from the *Vespa mandarinia* hornet in Japan. Mastoparan-M has the (INLKAIAALAKKLL) sequence. At a minimum concentration (MIC) of 0.5 nmol/mL, the peptide degranulated rat peritoneal mast cells [52]. Mast cell degranulation induced the release of inflammatory mediators, such as TNF-α, IL-1β, and nitrite, from cultured mouse spleen macrophages [53].

SPPS was used to synthesize D-mastoparan M (INLKAIALAKKLL) and L-mastoparan M (INLKAIAALAKKLL). D-mastoparan M showed MIC of 6.25 mg/L against *E. coli* and *Pseudomonas aeruginosa* and 3.12 mg/L against *S. aureus*. The antibacterial impact of D-mastoparan was twice as effective as L-mastoparan M. After the supplementation of D-mastoparan M, bacterial lysis was observed at 1 h and was completed after 4 h [54].

#### 2.2.2. Anoplin

Anoplin (ANP) is the smallest antimicrobial, naturally occurring peptide isolated from the solitary spider wasp *Anoplius samariensis* (Hymenoptera: Pompillidae) and contains ten amino acids (Table 1), making it an ideal research template [55]. The peptide causes mast cell degranulation and has antimicrobial activity [55]. The presence of four-polar residues makes ANP water-soluble. Its interaction with amphipathic environments, such as trifluoroacetic acid (TFE)/water mixtures, or with anisotropic media, such as sodium dodecyl sulfate (SDS) micelles or anionic vesicles, induces α-helical conformations and amphiphilic properties as indicated by the circular dichroism (CD) spectra [55]. ANP inhibited the proliferation of murine erythroleukemia (MEL) cells in a time- and dose-dependent manner. The IC_50_ values were 161.49, 121.03, and 114.88 μM at 24, 48, and 72 h, respectively. Disrupting the cell membrane integrity was the primary mechanism behind anoplin’s cytotoxicity [56].

Synthetic ANP peptides have a broad spectrum of antimicrobial activity against Gram-positive and Gram-negative bacteria. ANP antimicrobial activity is susceptible to salt. Gram-negative bacteria were entirely immune to ANP in high-salt media (150 mM NaCl); however, Gram-positive bacteria’s efficacy was greatly diminished [55]. Equally interesting, it stimulates rat peritoneal mast cell degranulation, and ANP’s hemolytic activity was relatively low or virtually inactive on human erythrocytes [55].

ANP’s activity is highly sensitive to minor changes of the primary structure, such as single amino acid mutations in certain positions. For example, 37 anoplin analogs have been synthesized by replacing single and multiple residues leading to a change in amphipathicity and charge. Accordingly, the effects against *S. aureus* and *E. coli* varied considerably depending on the hydrophobicity and position of the various replaced amines. Residue replacement at positions 5, and 8 with phenylalanine or tryptophan caused by an increase in antibacterial, and hemolytic activity owed to the role of these aromatic residues in the membrane anchoring. Lysine placement in position 8 improved peptide selectivity for prokaryotic cells due to the higher charge [57], whereas C-and N-terminal truncation and C-terminus deamidation drastically decreased peptide antibacterial properties [58]. Antimicrobial activities were measured against *E. coli* and *B. subtilis* for all three derivatives of ANP (ANP-NH_2_, D-ANP-NH_2_, and ANP-OH). Both amidated ANP derivatives display 50 μg/mL, MIC values for *B. subtilis*, and 100 μg/mL for *E. coli*. Alternatively, the deamidated form showed significantly lower bactericidal activity with MIC values of 200 μg/mL. The LD_50_ values for both amidated ANP forms were identical and approximately 10- to 30-fold lower than those of ANP-OH. ANP loses its biological activity after deamidation. Both amidated and carboxylated forms have secondary structures similar to those of the lytic ANP [59]. The natural cationic ANP was modified by substituting residues Gly1 for Lys or Arg, Arg^5^ for Phe, and Thr^8^ for Lys. The antimicrobial properties change dramatically, and high activity against Gram-negative bacterium *Zymomonas mobilis* at MIC 7 μg/mL was observed, compared to native peptide MIC 200 μg/mL; additionally, it was non-toxic to erythrocytes and resistant to proteolysis [10]. Interestingly, the antimicrobial activities of ANP and analogs ANP-2 (WLLKRWKLL-NH_2_), and ANP-4 (KLLKWKKLL-NH_2_) were significantly higher than ANP-1 (WLLKRWKKLL-NH_2_) and ANP-3 (KLLKWWKKLL-NH_2_). The highest antimicrobial activity against *B. subtili* was shown by ANP-2 and ANP-4 (MIC value: 4 μM) compared to the parent peptide (MIC value: 32 µM). ANP-4 treatment significantly reduced the mortality rate of mice infected with *E. coli* compared to ANP. ANP-4 is a novel analog of ANP with high antimicrobial activity and enzyme stability that represent it as a successful agent for infections treatment [60].

#### 2.2.3. Decoralin

Decoralin (Dec-NH_2_) is a peptide derived from the solitary Eumenine wasp (*Oreumenes decoratus*) [61] and was synthesized by solid-phase synthesis [62]. Equally important, a natural antimicrobial peptide, Dec-NH_2_, was isolated from wasp venom, and its synthetic derivatives were manufactured using peptide design. Dec-NH_2_ exhibits potent activity toward cancer cells at doses of 12.5 μmol/L and specific inhibition of MCF-7 breast cancer cells [62]. In a biological assessment, synthetic Dec-NH_2_ demonstrated strong broad-spectrum antimicrobial activity, slight mast cell degranulation, and leishmanicidal activity. The peptide displayed low hemolytic function against mouse erythrocytes, EC_50_ lower 300 µM [61]. A synthetic Dec-NH_2_ analog with *C*-terminal amidation demonstrated much more efficient activity against Gram-positive and Gram-negative bacteria and yeast. When isoleucine was substituted by phenylalanine residue at position 6, the peptide increased its resistance to degradation in bovine fetal serum. Besides that, lower hemolytic activity was obtained for [Pro]^4^-decoralin-NH_2_ and [Phe]^6^-Des[Thr]^11^-decoralin-NH_2_. The antimicrobial effect was increased in the case of [Phe]^9^-[Phe]^10^-Dec-NH_2_ (MIC = 0.39 vs. native peptide of 0.78 µmol/L) against *Micrococcus luteus* A 270 [63]. Torres et al. synthesized two leucine-substituted Dec-NH_2_ analogs; [Leu]^8^-Dec-NH_2_, and [Leu]^10^-Dec-NH_2_ using SPPS. [Leu]^10^-Dec-NH_2_ analog showed similar activity against *E. coli*, and *P. aeruginosa* (MIC 1.6 μmol/L), and higher activities against *M. luteus* and *C. albicans*. The same helical structure of the [Leu]^8^-Dec-NH_2_ analog exhibited evidential low activities against *M. luteus*, *E. coli*, *Salmonella arizonae*, *B. subtilis*, *P. aeruginosa*, and *C. albicans* [11]. The natural sequence of amidated Dec-NH_2_ and eight synthesized analogs, along with their biological activity toward *Plasmodium*, were reviewed. The Dec-NH_2_ template compound did not display antiplasmodial properties; on the other hand, it’s designed analogs showed significant antiplasmodial activity (>95%). The highest antiplasmodial behavior was achieved by mutations made to the N-terminus of Dec [64].

#### 2.2.4. Polybia-MP-I

Polybia-MP-I is one of the 14 amino acid residues (Table 1) of mastoparan peptides [65]. The peptide was derived from the venom of the social wasp *P. paulista*. It causes moderate mast cell degranulation, demonstrates chemotactic action for polymorphonuclear leukocytes, exhibits active antimicrobial activity, and is non-hemolytic to rat erythrocytes [66]. Polybia-MP-I is cytotoxic to leukemic T lymphocytes and strongly selective to these individual cells [67]. Polybia-MP-I has demonstrated antitumor action against bladder and prostate cancer [12]; however, this antitumor activity drastically decreased with the synthesized analogs (replacement of the amino acids at position 7, 8, or 9 with Pro residue). These substitutions influence the original helical structure and electrostatic equilibrium and increase the degree of peptide hydrophilic behavior (Pro^7^ and Pro^9^). Polybia-MP-I exerts pore formation and thus alters the intact cellular structure leading to a cytotoxic and antiproliferative outcome. It can selectively inhibit the proliferation of prostate cancer cell lines (PC-3), human bladder cancer cell lines (Biu87 and EJ), and human umbilical vein endothelial cell lines (HUVEC) at IC_50_ of 20.8, 25.32, and 36.97 μM, respectively [12].

The peptides polybia-MP-I (IDWKKLLDAAKQIL-NH_2_) and Asn-2-polybia-MP-I (INWKKLLDAAKQIL-NH_2_) were manually synthesized in the solid phase. Polybia-MP-I and N-2-polybia-MP-I exhibited a significant reduction in the pain threshold at 30, and 50 μg/50 μL as detected at 2, and 8 h after peptide injection into the hind paw of mice [68,69]. The polybia-MP-I analogs (proline replacement) showed reduced antibacterial activity compared to the parent. MIC values of polybia-MP-I were 4, 16, 16, and 32 μM for *B. subtilis*, *E. coli*, *S. epidermidis*, and *S. aureus*, respectively. The MIC value was 8 µM for *C. glabrata* versus 16 µM against *C. albicans*. The fungicidal activity of polybia-MP-I versus both *Candida glabrata* and *C. albicans* was measured as an minimum fungicidal concentration (MFC) of 32 μM [70,71]. 

#### 2.2.5. Polybia-CP

Polybia-CP has been isolated from *P. paulista*, and gradually synthesized by SPPS, and its effects on bacteria have been recorded [14]. Polybia-CP’s MICs for *E. coli*, *P. aeruginosa*, *S. aureus*, *S. epidemic*, and *B. subtilis* were 16, 128, 4, 16, and 4 μM, respectively, while the MBCs were 8, 16, 128, and 16 μM, for *B. subtilis*, *S. aureus*, and *E. coli*, respectively. The peptide was stable at different temperature ranges of 20–100 °C, and the temperature changes did not affect the MIC values [72]. Polybia-CP showed antimicrobial activities with MIC values of 4–64 μM in eight fungal strains, where the highest activity was noted against *C. tropicalis* at a MIC of 4 μM [13].

Synthetic Polybia-CP has potent antitumor activity against Biu87 and PC-3 cell lines. Cell proliferation inhibition was observed at IC_50_ of 17.84 and 11.01 μM, respectively. The cytotoxicity of polybia-CP was explained by the disruption of cell membrane integrity [14].

#### 2.2.6. Polydim-I

Polydim-I is a peptide derived from the venom of a neotropical wasp (*Polybia dimorpha*). The peptide contains 22 amino acid residues and is known for its amphipathic properties due to the presence of hydrophobic amino acid residues (e.g., methionine, leucine, valine, and proline) [15].

Polydim-I was synthesized with high quality (>99%), and the relevant peptide sequence was tested and validated by MS analysis. The synthetic peptide is active against *Mycobacterium abscessus* subsp. massiliense infections as described in in vitro and in vivo studies. In vitro study, the inhibition was 55 to 68% of *M. abscessus* subsp massiliense strains growth at a concentration of 15.2 μg/mL in which the cell shape was expressively damaged. The peptide prevents bacterial growth through the inhibition of protein synthesis, did not result in visible morphological changes. Polydim-I treatment at 2 mg/kg/mLW showed significant reduction of the bacterial load in in the lungs, spleen, and liver [15], and the antimicrobial properties against *S. aureus*, *E. coli*, *Enterococcus faecalis*, *Acinetobacter calcoaceticus-baumannii* were displayed with MIC_50_ values of 4.1, 50.7, 73.2, and 84.0 μg/mL, respectively [73].

#### 2.2.7. Protonectarina-MP and Agelaia-MP

Protonectarina-MP was isolated from *Protonectarina syleirae* venom and is a member of the 14 amino residue class of mastoparans [74]. The peptide protonectarina-MP-NH_2_ (INWKALLDAAKKVL-NH_2_) and its analogue protonectarina-MP-OH (INWKALLDAAKKVL-OH) were produced by step-by-step manual SPPS. Protonectarin-MP-NH_2_ is a powerful mast cell degranulating peptide with slightly higher degranulating activity (ED_50_ = 8 ×10^−5^ M) than the standard peptide (ED_50_ = 20 × 10^−5^ M). Protonectarina-MP-OH, and even at high concentrations, has reduced degranulation activity. Protonectarina-MP-NH_2_ has effective antimicrobial activity against both Gram-positive and Gram-negative bacteria, while protonectarina-MP-OH has much poorer antimicrobial activity [17].

Agelaia-MP is a mastoparan peptide that contains 14 residues (INWLKLGKAIIDAL-NH_2_) and is isolated from the venom of the social wasp *Agelaia pallipes*. It was characterized by its poor antimicrobial action and the lack of chemotaxis toward mast cells [74]. Using the Fmoc strategy, agelaia-MP has been chemically and manually synthesized. At a concentration of 10 μM, the peptide enhances the insulin secretion from the mice pancreatic islets using different glucose doses (2.8, 11.1, and 22.2 mM). In mouse models, agelaia-MP-I has a dose-dependent anti-nociceptive effect. For example, nociception significantly declined when the highest dosage (6.4 nmol) was administered, while the maximal effect was observed 4 h after the peptide injection [16].

Protonectin is derived from the venom of the neotropical social wasp (*Agelaia pallipes*), with a sequence of ILGTILGLLKGL-NH_2_. The peptide exhibits poor hemolysis to rat erythrocytes [74]. Protonectin has some mast cell degranulating activity and potent antimicrobial action with *E. coli*, *P. aeruginosa*, *B. subtilis*, and *S. aureus* at MICs of 25, 1.7, 3.1, and 12.5 µg/mL, respectively [74].

Protonectin and its three analogues were synthesized through a stepwise solid-phase assay by replacing L-proline. Proline is a unique amino acid among the 20 protein-forming amino acids because its amine nitrogen is linked to two groups of alkyls, making it a secondary amine. The insertion of proline inside the peptide considerably changes the secondary structure. Protonectin has demonstrated potent antibacterial action toward multidrug-resistant *S. aureus*, and *E. coli* at MICs of 8, and 32 μM, respectively. MBC values were 8, 8, 16, and 64 µM for *B. subtilis*, *S. epidermidis*, *S. aureus*, and *E. coli*, respectively, indicating potent bactericidal effect [75].

#### 2.2.8. Philanthotoxin-433 (PhTX-433)

Philanthotoxin-433 (PhTX-433) is a polyamine-based toxin isolated from Egyptian digger wasp (*Philanthus triangulum*) venom. The venom induces prey paralysis by suppressing nicotinic acetylcholine receptors (nAChRs) and ionotropic glutamate receptors (iGluRs). PhTX-433 is an important lead compound in neuropharmacology [76,77]. The action of 17 analogs of PhTX-343 against ganglionic (α3β4) and brain (α4β2) nAChRs has been expressed in Xenopus oocytes. IC_50_ values for PhTX-343 inhibition of α3β4 and α4β2 receptors were 7.7 and 80 nM, respectively [78]. Their total synthesis achieved good yield (77%) and purity (80%) using a mild borane reduction protocol of polyamide precursors to access the polyamine chains. The synthesis of PhTX-433 isomers proved this strategy’s potential for the generation of branched analogs [76].

## 3. Application of Wasp Venom and Their Nests in Nanotechnology

There is an increasing awareness of the use of metabolites of arthropods including insect for the green synthesis of nanoparticles [141]. In this context, Jalaei and colleagues published the first study on the green synthesis of gold nanoparticles (AuNPs) using *V. orientalis* wasp venom peptides. The AuNPs had been fabricated by adding 25 mL of stabilized SDS-NPs suspension (AuNP and SDS surfactant) to 750 μL of purified peptides (20 μM). The crystalline gold nanoparticles have been characterized ((TEM; average size 23.2 ± 2.7 nm), (XRD; average size 35 nm), (UV where the peak of AuNPs confirmed at 555 nm)), and FTIR data demonstrated that amide, amine, peptide, and protein compounds were involved in AuNP biosynthesis. Furthermore, wasp venom-mediated AuNPs depicted good antibacterial activity using microdilution assay against *E. coli*, *K. pneumonia*, *S. typhimurium*, *S. aureus*, *S. mutans*, and *B. cereus* with MIC values of 5.11, 10.08, 11.66, 28.04, 18.78, and 28.12 μg/mL, respectively, in comparison with the positive control (tetracycline, MIC = 0.98–2.21 μg/mL) and peptides (MIC = 11.77–59.66 μg/mL) [19].

Not only can wasp venom be utilized for the green synthesis of nanoparticles, but wasp nests can also be used for this application. In this context, Lateef and colleagues utilized paper wasp (*Polistes* sp.) nests to biofabricate silver nanoparticles by interacting 1 mL of nest extract to 40 mL of 1 mM of silver nitrate (AgNO_3_) solution. UV, FTIR, EDX, and TEM were used for the characterization of fabricated-AgNPs. These particles show wide applications like antimicrobial activity against *P. aeruginosa* and *K. granulomatis* (at concentrations of 60, 80, and 100 µg/mL), the particles 100 µg/mL show 100% inhibition for *Aspergillus flavus* and *Aspergillus niger*. Furthermore, at 20 and 40 µg/mL, malachite green degradation was reported to be 64.3%–93.1% within 24 h of application; they also exhibited excellent blood anticoagulation and blood clot dissolution [142]. Additionally, the mud dauber wasp-associated fungus, Talaromyces sp. (CMB-W045), could be used to recover Fe_2_O_3_ nanoparticles from natural extracts [20].

Since two decades ago, some peptides (CPPs; cell-penetrating peptides) have been used in drug delivery applications as nano-carriers for macromolecule delivery systems as they can penetrate living cells. Among CPPs, there is a chimeric peptide called transportan (TP), which consists of 27 amino acids derived from a fusion of the first 12 residues of N-terminus of neuropeptide galanin and lysine residues, thus binding the remaining 14 C-terminal residues of mastoparan, a wasp venom peptide [143]. Subsequently, TP with d-arginine was used to synthesize a 36 amino acid peptide called T9 (dR). T9 (dR) can condense siRNA into nanoparticles sized between 350 and 550 nm and is also used for siRNA delivery into different cells; and to inhibit the replication of the influenza virus [144]. A similar study showed that TP nanoparticles could be used to improve cutaneous paclitaxel delivery and three-dimensional (3D)-bioengineered skin against tumor cancer cell lines [21]. Regarding the application of peptides in fields like agriculture nanobiotechnology, many CPPs can translocate macromolecular cargo complexes of drugs, proteins, or nucleic acids that are much larger than the pores of plasma membranes inside plant cells, which can lead to a potential revolution in the field of cell biotechnology [143].

## 4. Wasp Venom, Allergies, and Sensitization: A Double-Edged Sword

Wasp venom may result in anaphylaxis, anaphylactic shock, or even death [145,146]. Compared to airborne allergens such as mites and pollen [147], wasp venom is injected into the skin, reaching the blood directly, and initiating a general immune response [145]. The most frequent clinical picture starts with swelling at the sting site, expands over a 10-cm-diameter, and lasts for longer than 24 h in approximately 15–20% of the general population [148]. Venom allergy can cause various reactions, categorized as common local, wide local, systemic anaphylactic, systemic toxic, and rare reactions [149]. Up to 3% of the population suffers from potentially either systematic, anaphylactic, or toxic reactions after wasp stings [150], which could be life-threatening and lead to sudden death. Approximately fifty fatalities from wasp stings are reported every year [151]. Phospholipase A1 (Ves v 1), hyaluronidase (Ves v 2), protease vitellogenin (Ves v 6), and antigen 5 (Ves v 5) are the most allergenic components of the genera Dolichovespula, Polistes, Vespula, and Vespa [152,153,154]. Phospholipase A1, hyaluronidase, and antigen 5 are the three main allergens described for *P. paulista* venom and cause a high number of allergic reactions and dozens of fatal anaphylaxis cases annually in Brazil [155,156,157].

Accurate diagnosis of venom allergy is largely based on history, skin testing, radioallergosorbent test (RAST), the identification of particular IgE (sIgE) antibodies and basophilic activation testing (BAT), and is thus contingent on the choice of allergen-directed immunotherapy (AIT) [158,159,160]. The skin test is considered the golden standard diagnostic test due to its sensitivity to different types of venom, its ability to give results within a few minutes and to simulate a sting’s real-life effects better than in vitro tests. Recombinant allergen-based BAT testing is a highly specific but sometimes less sensitive tool. The severity of sting reactions was inversely correlated with total IgE levels [161,162,163], negative skin test results and non-detectable IgE despite BAT positivity were seen in 73% of patients following an insect sting and were associated with a systemic allergic reaction. Component-resolved assessments are a beneficial addition to the diagnostic continuum as long as they are used in accordance with existing procedures. Commercially available recombinant allergens have modern processes operating, such as spiking yellow jacket venom (YJV) extract with Ves v 5, providing increased exposure to the initial allergen detectors [164]. Recently, rPoly p 1 (recombinant of phospholipase A1 (PLA1)) was known as marker of *P. paulista* allergy with high sensitivity of 95% [165].

Emergency medicines and venom immunotherapy (VIT) are two main therapeutic options to treat wasp venom allergy [145]. Emergency medicine refers to administering antihistamines, epinephrine (adrenaline), and corticosteroids to patients with wasp venom allergies. First-line treatment of VIT is recommended. VIT is extremely effective in enhancing patients’ quality of life and greatly decreasing the risk of systemic responses before and after therapy [145]. VIT includes venom injections with an up-dosing period over several weeks and a 3–5 year maintenance period. VIT is considered safe as most adverse events are mild and local during the induction and maintenance period [166]. However, a significant medication period (3–5 years) results in high costs and low adherence to the therapy. In order to overcome this hindrance, a clinical routine has been introduced with ultra-rush protocols for up-dosing in just a few hours. Ultra-rush therapy is safe, quick, and compliant, and therefore cost-effective [23,166,167].

Another major challenge facing VIT is cross-reactivity. Cross-reactivity can occur due to similarities in the content and composition of Vespula species of single allergen venom. Vespinae (Vespula, Vespa, and Dolichovespula) have high cross-reactivity with each other but are less cross-reactive than Polistes. Compared to American species, cross-reactivity between European Polistes species is generally very high [153]. Cross-reactivity among venoms of different species may be a testing challenge [168]. False or positive cross-reaction outcomes can occur when IgE is directed toward antigenic carbohydrate determinants [169]. Seven recombinant antigens 5 of the Vespoidea groups were engineered and tested to resolve immunological IgE cross-reactivity at the molecular level [170].

Therefore, the measurement of specific IgE (sIgE) antibodies plays a major role. Immunoglobulin E (IgE) antibodies are the main actors, causing histamine release, lipid mediators, enzymes, cytokines, and chemokines responsible for the acute allergic response [171]. Component-resolved diagnosis (CRD) has been established recently, allowing the measurement of sIgE antibodies against Ves v1, Ves v5, and Pol d5, as well as cross-reactive carbohydrate determinants (CCDs). Estimations of IgE antibodies to Ves v 1, in addition to Ves v 5, should be included in diagnostic work [172]. Ves v 5 and Pol d 5 are 23-kDa proteins present in the Vespidae family of venom hornet (*Vespula* spp.) and wasp (*Polistes dominulus*) venom. Both proteins are known as antigen 5 and are recognized as the largest and most active allergens [173]. These tests were designed to help evaluate the therapeutic validity of any given sensitization, especially in patients with dual sensitization.

Several studies have been conducted to compare the immunoprotective efficacy and safety of different wasp venom preparations. In 1986, and for the first time, Mosbech and colleagues investigated the allergen immunotherapy effectiveness of yellow jacket wasp venom with three different preparations using thirty-two yellow jacket venom-allergic patients. The preparations were YJ Pharmalgen (YJ venom reconstituted in albumin containing saline diluent), YJ venom extract adsorbed to aluminum hydroxide, and YJ venom extract non-adsorbed. The results demonstrated that the three extracts showed the same immunological effectiveness (the amount of specific IgG was 44.2 ± 3.8 µg/mL) except for some minor varieties representation in IgG response induction for a longer period treatment with aluminum hydroxide-adsorbed extract [174,175].

## 5. Concluding Remarks

A limited number of ingredients in venom have been defined despite the wide diversity of wasp species. Wasps can synthesize toxic compounds as a part of their prey control mechanism to capture prey and defend their colonies. The venomous components of predator wasps are a complex mixture of high molecular weight proteins (e.g., enzymes and allergens), small peptides, and low molecular weight compounds (e.g., bioactive amino acids and amines). The accessibility of extremely delicate techniques, such as mass spectrometry, enables a more effective assessment of proteomics and transcriptomics of a restricted number of venoms to understand their impacts. For use in the medical sector, these compounds are promising as each compound was described by its effects, such as antimicrobial, anticancer, histamine release, and anti-inflammatory ability. Accumulating functional information on venom component bioactivity would promote the use of wasp venoms for pharmacological and medical purposes. Moreover, synthesis protocols should be developed to provide a higher quantity of active compounds for further evaluation in order to be applied in the industry. Some issues hinder its effective application, such as allergies to wasp venom, and restricted extraction amount. Many attempts have been made to reduce venomous side effects using VIT, which acts as a basic treatment for patients allergic to hymenopteran venom. VIT improved patient quality of life by reducing the risk of systemic reactions during, and after treatment.

Considering the numerous studies that have been conducted on wasp venom little has been reported on the practical application of these compounds. Among the wasp components, the antimicrobial properties of mastoparan are the most tested and confirmed [41]. Moreover, several wasp venom proteins have been registered in drug banks for use in allergenic testing, including proteins from *Polistes metrics* [176], *Polistes exclamans* [177], *Vespula germanica* [178], *Polistes fuscatu* [179], *V. vulgaris* [180], *Dolichovespula arenaria* [181], *Dolichovespula maculate* [182], *Vespula pensylvanica* [183], and *Vespula maculifrons* [184].

Although wasp venom remains mostly unexplored, recent findings have shown that venom is also abundant in peptides, and these peptides are similar to other venomous species. Up to 134 venom peptides have been entirely sequenced to date (Table 1). The pharmacological properties of insect venom peptides are identified as antimicrobial.

## Figures and Tables

**Figure 1 toxins-13-00206-f001:**
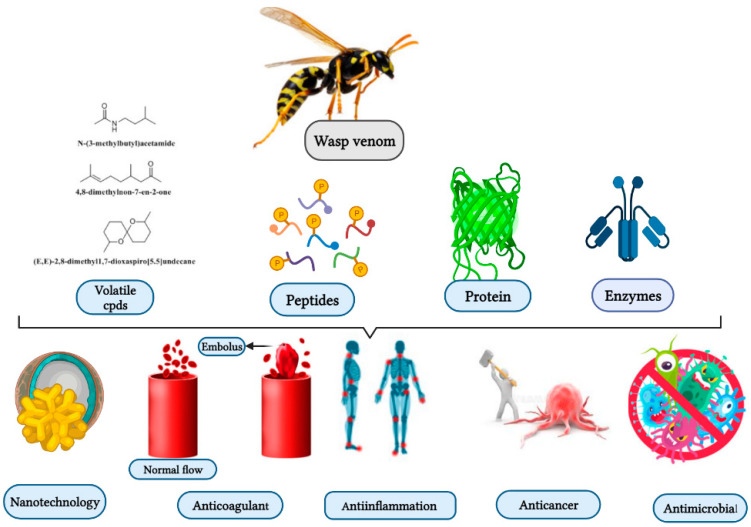
Wasp venom as a source of bioactive compounds and its biological activities and application.

**Figure 2 toxins-13-00206-f002:**
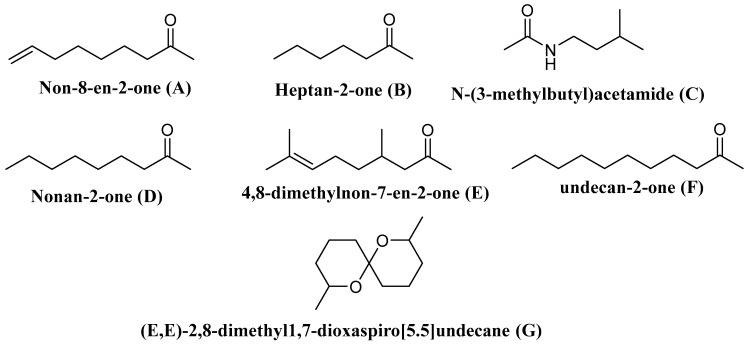
Some of the volatile compounds identified from wasp venom.

**Table 1 toxins-13-00206-t001:** Isolated constitutes from Wasp-Venom and their biological activity.

Wasp-Scientific Name	Isolated Compounds	Biological Activity	Reference
Peptides			
*Vespa xanthoptera* *Vespula lewisii*	Mastoparan (MPX) (INWKGIAAMAKKLL-NH_2_)	Cytotoxic against Glioblastoma multiforme (T98G) cell, 60% inhibition at 20 μmol/L (in vitro) Anti-*Escherichia coli* and anti-*Lactococcus lactis* at MIC 8, and 2.5 µM, respectively (in vitro).	[40,79,80]
*Anterhynchium flavomarginatum micado*	Mastoparan-AF (EMP-AF) (INLLKIAKGIIKSL-NH_2_)	Blocked lobster neuromuscular transmission. Mediated depolarization of the muscle membrane, often leading to a weak contraction of the muscle at 0.1 ± 1 mM (in vitro).	[1,81]
*V. lewisii, Vespa tropica* and *Polybia paulista*	Mastoparan (INLKALAALAKKIL)	Induces apoptosis in B16F10-Nex2 melanoma cells treated with 165 µM. Potent anti-inflammatory. Shows activity against colistin-susceptible *Acinetobacter baumannii* and colistin-resistant *Acinetobacter baumannii* at MIC_50_ value of 4, and 8 mg/l, respectively. Antimicrobial activity on the epimastigote, trypomastigote and amastigote forms of *Trypanosoma cruzi* Y strain via dose-dependent growth inhibition (in vitro).	[38,41,82]
*Vespa basalis*	Mastoparan B (LKLKSIVSWAKKVL)	Anti-*Enterococcus faecalis* and anti-*Bucillus subtilis* at MIC of 3.13 mg/mL (in vitro).	[51]
*V. basalis*	Mastoparan-I1 (INLKAIAALVKKVL)	ND	[51]
*V. basalis*	Mastoparan-A (IKWKAILDAVKKVI)	ND	[51]
*V. basalis*	Mastoparan-T (INLKAIAAFAKKLL)	ND	[51]
*Vespula vulgaris*	Mastoparan V1 (INWKKIKSIIKAAMN)	Potent antimicrobial activity against *Streptococcus mutans* and *Salmonella enterica* at 50 µM (in vitro).	[4]
*Vespa orientalis L.*	Mastoparan (HRI) (INLKAIAALVKKVL-NH_2_)	Cytotoxic towards T98G cells and give 80% inhibition at 20 μmol/L (in vitro).	[40]
*Vespa crabro*	Mastoparan-C (MP-C) (LNLKALLAVAKKIL-NH_2_)	Inhibition of the biofilm formation by *Staphylococcus Aureus* and *Pseudomonas aeruginosa* at 32 μM MBIC (in vitro).	[26]
*V. tropica*	Mastoparan-VT1 (INLKAIAALAKKLL)	Anti-*E. faecalis* at 2.5 µg/mL (in vitro).	[30]
*V. tropica*	Mastoparan-VT2 (NLKAIAALAKKLL)	Anti-*E. faecalis*, anti-*E.coli* and anti-*S.aureus* at 5 µg/mL (in vitro).	[30]
*V. tropica*	Mastoparan-VT3 (INLKAITALAKKLL)	*Anti-S. aureus* and anti-*Candida parapsilosis* at 2.5 µg/mL (in vitro).	[30]
*V. tropica*	Mastoparan-VT4 (INLKAIAPLAKKLL)	Anti-*Bacillus pyocyaneus*, anti-*P. aeruginosa*, and anti-*Bacillus dysenteriae* at 10 µg/mL (in vitro).	[30]
*V. tropica*	Mastoparan-VT5 (VIVKAIATLASKLL)	Anti-*Candida albicans* at 40 µg/mL (in vitro).	[30]
*V.tropica*	Mastoparan-VT6 (INLKAIAALVKKLL)	Anti-*S. aureus* and *anti-B. dysenteriae* at 20 µg/mL (in vitro).	[30]
*V. tropica*	Mastoparan-VT7 (INLKAIAALARNY)	Anti-*E. faecalis* at 5 µg/mL (in vitro).	[30]
*Polistes rothneyi iwatai*	Polistes-mastoparan-R1 (Pm-R1) (INWLKLGKKILGAI-NH_2_)	Has histamine-releasing activities from rat mast cells (EC_50_ = 0.09 µM) (in vitro).	[80]
*P. rothneyi iwatai.*	Polistes-mastoparan-R3 (Pm-R3) (INWLKLGKQILGAL-NH_2_)	Has histamine-releasing activities from rat mast cells (EC_50_ = 0.19 mM) (in vitro).	[80]
*Vespa magnifica*	Peptide 5e (FLPIIAKLLGLL)	Anti-*S. aureus*, MIC = 5 µg/mL (in vitro).	[83]
*V. magnifica*	Peptide 5f (FLPIPRPILLGLL)	Anti-*S. aureus*, MIC = 10 µg/mL (in vitro).	[83]
*V. magnifica*	Peptide 5g (FLIIRRPIVLGLL)	Anti-*S. aureus* MIC = 10 µg/mL (in vitro).	[83]
*V. magnifica*	Peptide 12a (INWKGIAAMAKKLL)	Anti-*S. Aureus,* and anti-*C. albicans* at MIC = 3.7 µg/mL (in vitro).	[83]
*V. magnifica*	Peptide 12b (INWKGIAAMKKLL)	Anti-*S. aureus* MIC = 3.7 µg/mL (in vitro).	[83]
*P. dimorpha*	Polydim-I (AVAGEKLWLLPHLLKMLLTPTP)	Antimycobacterial activity at 7.6 μg/mL (in vitro). Anti-*S. aureus* at MIC_50_ 4.1 µg/mL (in vitro).	[15,73]
*Anoplus samariensis*	As-126 (EDPPVVKMK-NH_2_)	ND	[84]
*Batozonellus maculifrons*	Bm-10 (ETAPVPKAISK-NH_2_)	ND	[84]
*A. samariensis*	Anoplin (GLLKRIKTLL-NH_2_)	Cytotoxic for T98G cells, gives 10% inhibition at 20 μmol/L (in vitro).	[40,55]
*P. hypochondriaca*	Pimplin (KRKPPRPNPKPKPIP)	Effective against *Musca domestica* at dose of 40 ng (in vitro).	[85]
*A. flavomarginatum* micado	Af-113 (INLLKIAKGIIKSLNH_2_)	ND	[86]
*Agelaia vicina*	Agelaiatoxin-8 (AVTx8) (INWKLGKALNALLNH_2_)	Inhibits gamma-aminobutyric acid (GABA) neurotransmission uptake at EC_50_ value of 0.09 ± 0.04 µM and maximum inhibition of 97 ± 5% (in vitro).	[87]
*Agelaia pallipes* pallipes	AgelaiaMP-I (INWLKLGKAIIDAL-NH_2_)	Has hemolytic activity at ED_50_ = 60 µM.	[28]
*A. pallipes* pallipes	AgelaiaMP-II (INWKAILQRIKKML-NH_2_)	Has hemolytic activity at ED_50_ = 240 µM (in vitro).	[88]
*Anoplius samariensis*, and *Batozonellus* *maculifrons*	Pompilidotoxins (α-PMTXs) (RIKIGLFDQLSKL-NH_2_)	Facilitates synaptic transfer in the motor neuron of the lobster and delays downregulation of the sodium channel (in vitro).	[89]
*A. samariensis*, and *B.* *maculifrons*	β-PMTXs (RIKIGLFDQRSKL-NH_2_)	Facilitates synaptic transfer in the neuromuscular junction of the lobster, and slows the sodium channel inactivation (in vitro).	[89]
*A. flavomarginatum micado*	Eumenine mastoparan-AF (EMP-AF) (INLLKIAKGIIKSL-NH_2_)	Effective hemolytic response in human erythrocytes. Enhancing degranulation of rat peritoneal mast cells and RBL-2H3 cells (in vitro).	[81]
*Agelaia pallipes pallipes,* and *Protonectarina sylveirae*	Protonectin (ILGTILGLLKGL-NH_2_)	Antibacterial activity towards Gram-positive and Gram-negative bacteria. Releasing Lactate dehydrogenase (LDH) from mast cells. Chemotaxis against polymorphonuclear leukocytes (PMNL) (in vitro).	([90]
*A. pallipes pallipes,* and *P. sylveirae*	Protonectin (1–6) (ILGTIL-NH_2_)	ND	[90]
*A. pallipes pallipes*	Protonectin (1–4)-OH (ILGT-OH)	Has poor hemolytic activity at ED_50_ = 1 mM (in vitro).	[88]
*A. pallipes pallipes*	Protonectin (7–12) (GLLKGL-NH_2_)	ND	[88]
*A. pallipes pallipes*	Protonectin (1–5)-OH (ILGTI-OH)	Has weak hemolytic activity at ED_50_ = 1 mM (in vitro).	[88]
*A. pallipes pallipes*	Protonectin (1–6)-OH (ILGTIL-OH)	Has poor hemolytic activity at ED_50_ = 1 mM (in vitro).	[88]
*Orancistrocerus drewseni*	Orancis-protonectin (ILGIITSLLKSL-NH_2_)	Has hemolytic activity of the sheep blood cells at 50 µM (in vitro).	[91]
*A. pallipes pallipes*	Pallipine-I (GIIDDQQCKKKPGQSSPVCS-OH)	ND	[88]
*A. pallipes pallipes*	Pallipine-II (SIKHKICKLLERTLKLTT PFC-NH_2_)	ND	[88]
*A. pallipes pallipes*	Pallipine-III (SIKKHKCIALLERRGGSKLPFC-NH_2_)	ND	[88]
*P. paulista*	Paulistine (SIKDKICKIIQCGKKLPFT-NH_2_) (oxidized form)	Causes mast cells degranulation or hemolysis (in vitro).	[92]
*Vespa mandarinia*	Ves-CP-M (FLPILGKLLSGL-NH_2_)	ND	[65]
*V. xanthoptera*	Ves-CP-X (FLPIIAKLLGGLL)	ND	[65]
*Paravespula lewisi*	Ves-CP-P (FLPIIAKLVSGLL)	ND	[65]
*V. tropica*	Ves-CP-T (FLPILGKILGGLL)	ND	[65]
*V. crabro*	Crabrolin (FLPLILRKIVTAL-NH_2_)	Releases histamine from rat peritoneal mast cells at ED_50_ of 11.8 µg/mL (in vitro).	[93,94]
*Eumenes rubronotatus*	Eumenitin (LNLKGIFKKVASLLT)	Shows antimicrobial activity against *S. aureus*, *Staphylococcus saprophytius*, *E. coli* at MIC = 6 µM (in vitro).	[95]
*E. rubrofemoratus*	Eumenine mastoparan-ER (EMP-ER) (FDIMGLIKKVAGAL-NH_2_)	Anti-*C. albicans* at MIC 7.5 µM. Has Leishmanicidal activity at IC_50_ 20 µM (in vitro).	[96]
*Eumenes micado*	Eumenine mastoparan-EM1 (LKLMGIVKKVLGAL-NH_2_)	Anti-*S. aureus* and anti-*E. coli* at MIC 7 µM (in vitro). Has Leishmanicidal activity with an IC_50_ of 36 µM (in vitro).	[97]
*E. micado*	Eumenine mastoparan-EM2 (LKLLGIVKKVLGAI-NH_2_)	Anti-*S. aureus* and anti-*E. coli* at MIC of 3 µM (in vitro). Has Leishmanicidal activity with an IC_50_ of 36 µM (in vitro).	[97]
*Eumenes fraterculus*	Eumenine mastoparan-EF (EMP-EF) (FDVMGIIKKIASALNH_2_	Anti-*C. albicans* at MIC of 7.5 µM. Has Leishmanicidal behavior at IC_50_ of 40 µM (in vitro).	[96]
*O. drewseni*	Eumenine mastoparan-OD (EMP-OD) (GRILSFIKGLAEHL-NH_2_)	Induces hemolysis of the sheep blood cells at 50 µM (in vitro).	[91]
*E. rubrofemoratus*	Eumenitin-R (LNLKGLIKKVASLLN)	Anti-*Sreptococcus pyogenes*, anti-*Micrococcus luteus*, and anti-*Stenotrophomonas maltophilia* at MIC of 15 µM. Anti-*B. subtilis* at MIC 7.5 µM (in vitro).	[96]
*E. fraterculus*	Eumenitin-F (LNLKGLFKKVASLLT)	Anti-*C. albicans* at MIC of 7.5 µM. Has Leishmanicidal activity at IC_50_ of 52 µM (in vitro). Anti-*S. maltophilia* at MIC of 15 µM (in vitro).	[96]
*P. paulista.*	Polybia-CP (ILGTILGLLKSL-NH_2_)	Anti-microbial against *S. aureus* and *B. subtilis* at 15 µg/mL compared with 0.5 and 18 µg/mL of tetracycline (in vitro).	[14,65]
*P. paulista*	Polybia-CP 2 (ILGTILGKIL-OH)	Has chemotaxis, mast cell degranulation, and hemolytic activities (*in vivo*).	[98]
	Polybia-CP 3 (ILGTILGTFKSL-NH_2_)	Has chemotaxis, mast cell degranulation, and hemolytic activities (*in vivo*). Antiplasmodial and anticancer properties (in vitro).	[8,98]
*P. paulista*	Polybia-MP1 (IDWKKLLDAAKQIL-NH_2_)	Antitumor against bladder and prostate cancer cells. Exhibits potent activity against *S. aureus*, MIC of 9 µΜ (in vitro). Anti-*C. albicans* (EC_50_ = 12.9 μM) and *C. neoformans* (EC_50_ = 11 μM) (in vitro). Fungicidal activity against *Candida glabrata* (EC_50_ = 8 μM) and *C. albicans* (EC_50_ = 16 μM) (in vitro). Anti-*E. coli*, *P. aeruginosa, B. subtilis,* and *S. aureus* at MIC of 8, 8, 4, and 15 μg/mL compared to 2, 18, 18, and 0.5 of tetracycline (in vitro).	[64,85]
*V. orientalis* L.	HR-1 (INLKAIAALVKKVL-NH_2_	ND	[99]
*V. orientalis* L.	HR-2 (FLPLILGKLVKGLL-NH_2_)	ND	[99]
*Polistes jadwigae*	Polisteskinin-J (RRRPPGFSPFR-OH)	ND	[98]
*Pollistes chiensis*	Polisteskinin-C (SKRPPGFSPFR-OH)	ND	[98]
*P. rothney*	Polisteskinin-R (ARRPPGFTPFR-OH)	Exerts potent anxiolytic effects at 6, 3, and 1.5 ηmol compared to positive control Diazepam (in vivo)	[98,100]
*Vespa analis*	Vespakinin-A (GRPPGFSPFRVI-OH)	ND	[98]
*Vespa mandarínia*	Vespakinin-X (ARPPGFSPFR-OH)	ND	[98]
*V. magnifica, Parapolybia varia, V. tropica*	Vespid Chemotactic Peptides (VCP)	Anti-tumor activities towards NIH-OVCAR-3 and SK-OV-3 ovarian cancer cell lines at concentrations higher than 10 μM (in vitro).	[34,101]
*V. magnifica (Smith)*	VCP-5h (FLPIIGKLLSGLL-NH_2_)	MICs of 5, 25, and 30, µg/mL for *S. aureus, C. albicans* and *E. coli*, respectively (in vitro).	[102]
*Parapolybia varia*	Vespakinin (Vespk)	Antitumor activity to SK-OV-3 at 24 h post-treatment (in vitro).	[101]
*V. magnifica*	Vespakinin-M GRPPGFSPFRID	ND	[103]
*Batozonellus maculifrons*	Pompilidotoxins (β-PMTXs) (RIKIGLFDQLSRL-NH_2_)	Inactivation of the Na^+^ channel, and the Nav1.6 channel was more selective (in vitro).	[1]
*O. drewseni*	OdVP1 (GRILSFIKGLAEHL-NH_2_)	Anti-*E. coli,* and anti-*C. albicans* at MIC of 6 µM (in vitro).	[104,105]
*O. drewseni*	OdVP2 (ILGIITSLLKSL-NH_2_)	Anti-*S. aureus* at MIC of 25 µg/mL. Anti-gray mold *Botrytis cinerea* at MIC of 0.4 µM (in vitro).	[104,105]
*O. drewseni*	OdVP3 (KDLHTVVSAILQAL-NH_2_)	Anti-gray mold *B. cinerea* at MIC of 5 µM (in vitro).	[104,105]
*O. drewseni*	OdVP4 (LDPKVVQSLL-NH_2_)	ND	[104]
*Nasonia vitripennis*	Defensin-NV (VTCELLMFGGVVGDSACAANCLSMGKAGGSCNGGLCDCRKTTFKELWDKRFG)	Anti-*S. aureus*, and Anti-*B. cereus* at MIC of 0.93 µM (in vitro). Anti-*B. dysenteriae* at MIC of 0.46 µM (in vitro). Anti-*E. coli*, and anti-*C. albicans* at MIC of 1.86 µM (in vitro). Anti-*P. aeruginosa* at MIC of 9.3 µM (in vitro).	[106]
*Chartergellus communis*	Communis (INWKAILGKIGK-COOH)	ND	[107]
*C. communis*	Communis-AAAA (INWKAILGKIGKAAAAVNH_2_)	Hemolytic activity at EC_50_ = 142.6 μM (in vitro). Hyperalgesic effect at 2 nmol/animal (in vivo).	[107]
*Cyphononyx* *Fulvognathus*	Bradykinin (RPPGFSPFR)	Acts as a chemoattractant directing glioma cells into blood vessels in the brain of rats (in vivo).	[108]
*Megascolia flavifrons*, and *Colpa interrupta*	Megascoliakinin = Thr6BK-Lys-Ala (BK = bradykinin) (RPPGFTPFRKA)	Prevents the synaptic transmission of the nicotinic acetylcholine receptor (nAChR) in the central nervous system of insect (in vitro).	[109]
*C. fulvognathus and* *P. paulista*	RA-Thr6 -Bradykinin (RARPPGFTPFR-OH)	ND	[98]
*Polybia occidentalis, M. flavifrons, C. interrupta,* and *P. paulista*	Threonine6-bradykinin (Thr6-BK) RPPGFTPFR-OH	Anti-nociceptive effects with approximately two-fold higher than bradykinin and morphine (in vivo).	[98,110]
*P. paulista*	RA-Thr6 -Bradykinin-DT (RARPPGFTPFRDT-OH)	ND	[98]
*C. fulvognathus*	Fulvonin (SIVLRGKAPFR)	Displays hyperalgesic impact after intraplantar injection in the rat paw pressure test (in vivo).	[111]
*C. fulvognathus* (Japan)	Cyphokinin (DTRPPGFTPFR)	Demonstrates hyperalgesic impact after intraplantar injection in the rat paw pressure test (in vivo).	[111]
*C. fulvognathus* (Japan)	Cd-146 (SETGNTVTVKGFSPLR)	Shows hyperalgesic effect in the rat paw pressure test after intraplantar injection (in vivo).	[111]
*C. fulvognathus*	Cd-125 (DTARLKWH)	ND	[111]
*P. paulista*	Mastoparan (MPI) (IDWKKLLDAAKQIL-NH_2_)	Cytotoxic towards T98G cells, gives 30% inhibition at 20 μmol/L (in vitro).	[40]
*Pseudopolybia vespiceps*	Mastoparan Polybia-MPII (INWLKLGKMVIDAL-NH_2_)	Anti-*staphylococcal* activity with an EC_50_ of 1.83 μM and EC_90_ of 2.90 μM (in vitro). Mice treated with 5 mg/kg showed a decline in bacterial load from 108 to ca. 106 CFUs (in vitro). Potent hemolytic activity against mouse cells (EC_50_ = 24.18 Μm, EC_90_ = 58.12 μM) (in vitro). Inhibits the growth of *C. neoformans* (EC_50_ = 11 μM) and *C. albicans* (EC_50_ = 12.9 μM) (in vitro). Anti-*A. baumannii* AB 0 at MIC of 12.5 µM while MIC against *A. baumannii* AB 53 and AB 72 was 6.25 µM (in vitro). Adhesion inhibition for *A. baumannii* AB 02 and AB 72 at 25 µM while *A. baumannii* AB 53 was inhibited at a concentration of 12.5 µM (in vitro).	[28,112]
*P. paulista*	Polybia-MPIII (INWLKLGKAVIDAL)	Anti-*S. aureus*, MIC of 19 μM (in vitro).	[65]
*P. paulista*	Polybia-MP IV (IDWLKLRVISVIDL-NH_2_)	Shows strong mast cell degranulation. Has weak haemolytic activity, hypernociception and edema formation (in vitro).	[98]
*P. paulista*	Polybia-MP V (INWHDIAIKNIDAL-NH_2_)	Medium mast cell degranulation, haemolytic activity and hypernociception (in vitro).	[98]
*P. paulista*	Polybia-MP VI (IDWLKLGKMVM-OH)	Medium haemolytic activity and hypernociception (in vitro).	[98]
*P. paulista*	unk-1 (IPAGWAIVKV-NH_2_)	Shows weak mast cell degranulation and haemolytic activity (in vitro).	[98]
*P. paulista*	unk-2 (TGDSPDVR-OH)	Shows weak mast cell degranulation and haemolytic activity, weak chemotaxis for PMNLs, and a range of weak to strong hypernociception and oedema formation (in vitro).	[98]
*V. orientalis L.*	Orientotoxin (Neurotoxin)	Has lysophospholipase activity and inhibits both mediated and spontaneous release of the neurotransmitter from the presynaptic nerve membrane (in vivo).	[113,114]
*V. orientalis* L.	Peptide I (AGVILFGR-NH_2_)	Histamine release from mast cells ED_50_ = 5.10^−7^ (in vivo).	[115]
*V. orientalis* L.	Peptide II (AGVIFRSP-NH_2_)	Histamine release from mast cells ED_50_ = 3.10^−6^ (in vivo).	[115]
*Oreumenes* *decoratus*	Decoralin (De-NH_2_) (SLLSLIRKLIT-NH_2_)	Has hemolytic activity at EC_50_ of 80 µM (in vitro). Anti-*S. aureus*, MIC = 4 µM (in vitro). Anti-*B. Subtilis*, MIC = 8 µM (in vitro). Anti-*C. albicans*, MIC = 20 µM (in vitro). Has leishmanicidal activity, IC_50_ =11 µM (in vitro).	[61]
*V. ducalis*	VACP1 (AQKWLKYWKADKVKGFGRKIKKIWFG)	Potently inhibits cell proliferation and promotes the cell apoptosis of osteosarcoma (OS) cells, and this was concomitant with the activation of the JNK and p38 MAPK signaling pathway (in vitro).	[6]
*Emerald Jewel*, and *Ampulex compressa*	Ampulexin-1 (axn1) (CKDDYVNPKEQLGYDILEKLRQKP)	ND	[116]
	Ampulexin -2 (axn2) (CQNDYVNPKLQFACDLLQKAKERQ)	ND	[116]
	Ampulexin -3 axn3 SFSMLLQKAKERQ	ND	[116]
*V. orientalis*	AuNPs+ peptide (INLKAIAALVKKV)	Antibacterial using AuNPs against *K. pneumoniae, B. cereus, S. mutans*, *S. typhimuriu, E. coli,* and *S. aureus*, and with the inhibition zones of 9.21, 14.32, 14.71,19.21, 15.24 and 15.33 mm, respectively (in vitro).	[19]
*Vespa bicolor* Fabricius	*V. chemotatic* peptide (VESP-VBs) (FMPIIGRLMSGSL)	Anti-*S. aureus*, MIC = 1 µg/mL (in vitro).	[5]
*V. bicolor* Fabricius	*V. mastoparan* (MP-VBs) (INMKASAAVAKKLL)	Anti-*S. aureus*, MIC = 1.9 µg/mL (in vitro).	[5]
*Polistes dominulus*	Dominulin A (INWKKIAEVGGKILSSL)	Anti-*B. Subtilis,* and *E. coli* at MIC = 2 and 8 µg/mL, respectively (in vitro).	[117]
*P. dominulus*	Dominulin B (INWKKIAEIGKQVLSAL)	Anti-*B. Subtilis,* and *E. coli* at MIC = 2 and 8 µg/mL, respectively (in vitro).	[17]
*Protonectarina sylveirae*	Protonectarina-MP (INWKALLDAAKKVL)	Anti-*B. subtilis* and anti-*S. Aureus* MIC = 3.9 µg/mL (in vitro).	[69]
*Parapolybia indica*	Parapolybia-MP (INWKKMAATALKMI-NH_2_)	Anti-*S. aureus*, MIC = 3.9 µg/mL (in vitro).	[69]
*P. jadwigae*	Polistes mastoparan (VDWKKIGQHIKSVL)	Degranulation of mast cells at 5 nM/mL.	[39]
*V. magnifica* (Smith)	Vespid chemotactic peptide (VCP)	MICs for *S. aureus, C. albicans,* and *E. coli* were 5, 25, and 30, µg/mL, respectively (in vitro).	[102]
*V. bicolor* Fabricius	VESP-VB1 (FMPIIGRLMSGSL)	Anti-*E. coli*, MIC = 7.5 µg/mL (in vitro). Anti-*S. aureus*, MIC = 1.9 µg/mL (in vitro). Anti-*P. aeruginosa*, MIC = 3.75 µg/mL (in vitro). Anti-*C. albicans*, MIC = 30 µg/mL (in vitro).	[5]
*V. bicolor* Fabricius	MP-VB1 (INMKASAAVAKKLL)	Anti-*E. coli*, MIC = 15 µg/mL (in vitro). Anti-*S. aureus*, MIC = 3.75 µg/mL (in vitro). Anti-*P. aeruginosa*, MIC = 15 µg/mL (in vitro). Anti-*C. albicans*, MIC = 15 µg/mL (in vitro).	[5]
*V. tropica*	VCP-VT1	Anti-*E. coli*, *Enterobacter cloacae*, and *C. parapsilosis* at 2.5 µg/mL and Anti-*S. aureus* at 1.2 µg/mL (in vitro).	[30]
*V. tropica*	VCP-VT2 FLPIIGKLLSG	Antimicrobial against *S. aureus, E. cloacae* at 2.5 µg/mL (in vitro).	[30]
*Protopolybia exigua (Kinins)*	Protopolybiakinin-I (DKNKKPIRVGGRRPPGFTR-OH)	Caused degranulation of 35% of the mast cells (in vitro).	[118]
*P. exigua*	Protopolybiakinin-II (Kinins) (DKNKKPIWMAGFPGFTPIR-OH)	Caused degranulation of 52 % of the mast cells (in vitro).	[118]
*V. mandarinia*	VESCP-M2 (FLPILAKILGGLL)	Induces pain and severe tissue injury, oedema, cutaneous necrosis, and blister.	[119]
*Polistes lanio lanio*	PllTkP-I (QPPTPPEHRFPGLM)	ND	[120]
*P. lanio lanio*	PllTkP-II (ASEPTALGLPRIFPGLM)	ND	[120]
*V. magnifica (Smith)*	5-Hydroxytryptamine	ND	[121]
*V. magnifica (Smith)*	Vespakinin-M (GRPPGFSPFRID-NH_2_)	ND	[121]
*V. magnifica (Smith)*	Mastoparan M (INLKAIAALAKKLL-NH_2_)	ND	[121]
*V. magnifica (Smith)*	Vespid chemotactic peptide M (FLPIIGKLLSGLL-NH_2_)	ND	[121]
*Sphex argentatus* argentatus	Sa12b (EDVDHVFLRF)	Inhibits acid-sensing ion channels (ASIC) of rat dorsal root ganglion (DRG) neurons at IC_50_ of 81 nM while inhibiting it completely at 1 μM (in vivo).	[122]
*Isodontia harmandi*	Sh5b(DVDHVFLRF-NH_2_)	ND	[122]
*P. paulista*	Neuropolybin	Antiseizure	[37]
*Synoeca surinama*	Synoeca-MP I/LNWI/LKI/LGKKI/LI/LASL/NH_2_	Antimicrobial activity, MIC_50_ values were 1.9, 2, 8.3, 5.2, and 3.5 μM for methicillin-resistant *S. aureus*—MRSA, *E. coli* ESBL, vancomycin-resistant *E. Faecalis*, *P. aeruginosa* metallo-ß-lactamase, and *Klebsiella pneumoniae* KPC, respectively (in vitro). Anti-Candida species, with MICs varying from 10–40 μM (in vitro).	[123]
*Enzymes and proteins*			
*V. magnifica*	Magnifin (PLA1)	Activates platelet aggregation and induces thrombosis at 18 nM with causes 85% washed platelets aggregation in 60 s (in vivo).	[124]
*P. paulista* *(southeast Brazil)*	Phospholipase A1(Ves v 1)	Catalyzes the ester bonds hydrolysis of 1,2-diacyl-3 snglycerophospholipids at the sn-1 and sn-2 positions, respectively.	[125]
*P.paulista*	Phospholipase A_1_	Hydrolyzes phospholipids and produces 2-acyl-lysophospholipids and fatty acids.	[125,126]
*P. Occidentalis* and *P. paulista*	Phospholipase A2 (PLA2)	Potent hemolytic actions in washed red cells (in vitro). Hydrolyzes natural phospholipids, catalysing the deacylation of 1,2-diacyl-sn-3-phosphoglycerides at position 2 and thus releases free fatty acids and lysophospholipids (in vitro).	[127,128]
*P. paulista, Vespula maculate, Vespula arenaria, V. crabro, V. orientalis,* *Paravespula germanica,* *Paravespula vulgaris, Dolichovespula saxonica, Dolichovespula media,* and *Polistes* *Gallicus*	Hyaluronidase (Polyp2)	Hydrolyses hyaluronic acid which facilitates the diffusion of toxin into the tissue and blood circulation of the prey.	[129,130,131]
*Polistes comanchus*	Polistin (protein)	Responsible for the cytotoxic effect of the whole venom.	[132]
*P. paulista*	Antigen5 (Polyp5)	Major allergen could be used for allergy diagnostics and treatment.	[133]
*Cyphononyx dorsalis*	Arginine kinase-like protein	Exhibits paralytic activity in spiders with the same characteristic symptoms as the crude venom.	[134]
*Pteromalus puparum*	Vn.11 (protein)	ND	[135]
*Cotesia rubecula*	Vn 4.6	ND	[136]
*V. magnifica*	Magnvesin	Exerts anti-coagulant properties via hydrolyzing coagulant factors VII, VIII, TF, IX and X.	[137]
Some volatile compounds			
*Vespa velutina*	Undecan-2-one	Elicits the defense behavior	[138]
*V. velutina*	Non-8-en-2-one		
*V. velutina*	Nonan-2-one		
*V. velutina*	Heptan-2-one		
*V. velutina*	4,8-Dimethylnon-7-en-2-one		
*Polistes metricus* Say, *Polistes bellicosus* Cresson, and *Polistes dorsalis* (F.), as well as workers of *Polistes aurifer* (Saussure), *P. bellicosus, P. metricus*, and *P. dorsalis*	*N-*(3-Methylbutyl)acetamide	ND	[139]
*P. occidentalis*	(E,E)-2,8-Dimethyl1,7-dioxaspiro[5.5]undecane	Elicit the defense behavior	[140]

ND: Not detected.

## Data Availability

No new data were created or analyzed in this study. Data sharing is not applicable to this article.

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
