# Peer review of "Wasp Venom Biochemical Components and Their Potential in Biological Applications and Nanotechnological Interventions"

_toxins, 2021, doi:10.3390/toxins13030206_

Round 1

Reviewer 1 Report

This is a comprehensive and well-written review on the biochemical properties of venom components and their potential use as antimicrobial, anticancer, and anti-inflammatory compounds. To the best of my knowledge, this review also covers a new and interesting aspect of the application of wasp venom components in nanotechnology. The authors went to a great effort to organize an exhaustive list of the Isolated components and their biological activity with bibliographic sources from pages 8 to 16 (table 1). Personally, I learned new things and I am sure this review is timely and it should raise a strong interest in the “wasp venom” community.

Author Response

Reviewer (1) comments

This is a comprehensive and well-written review on the biochemical properties of venom components and their potential use as antimicrobial, anticancer, and anti-inflammatory compounds. To the best of my knowledge, this review also covers a new and interesting aspect of the application of wasp venom components in nanotechnology. The authors went to a great effort to organize an exhaustive list of the Isolated components and their biological activity with bibliographic sources from pages 8 to 16 (table 1). Personally, I learned new things and I am sure this review is timely and it should raise a strong interest in the “wasp venom” community.

Response:

We would like to thank the reviewer for the nice words and encouragement.

Reviewer 2 Report

Wasp venom: A systematic review of its biochemical properties, biological studies, and potential nanotechnology applications

In the present review, the authors highlighted the medicinal value of the wasp venom compounds, as well as limitations and possibilities. In my opinion, the study is still interesting and innovative, including was well delineated. However, I have some comments:

Comment (1): In my opinion, title is good but it should be changed to be more attractive.

Comment (2): General proofreading is highly recommended.

Comment (3): Introduction. There is a brief review of existing knowledge and relevance of study.

Comment (4): I recommend to the authors to merge the two sub-titles, 2. Biological properties of wasp venom and 3. Isolated and synthesized bioactive peptides from wasp venom, together because they give the same meaning and can be one section.

Comment (5): Figures and legends. Legends are adequate and figures are necessary to understand the results obtained. Authors need to make a space between the compounds in figure 1 and mark each one by A, B, C, …..

Author Response

Dear reviewer

We thank you for the valuable comments regarding our article entitled, "Wasp venom: A systematic review of its biochemical properties, biological studies, and potential nanotechnology applications".

We agree with all comments and found them very helpful; we would like to thank you for taking the time and efforts necessary to provide such insightful guidance.

We address each criticism and the comments made individually, and explain how we have modified the manuscript to address the concerns that were expressed. We kindly ask that you consider the revised article for publication by Toxins journal.

Reviewer (2) comments

Wasp venom: A systematic review of its biochemical properties, biological studies, and potential nanotechnology applications.

In the present review, the authors highlighted the medicinal value of the wasp venom compounds, as well as limitations and possibilities. In my opinion, the study is still interesting and innovative, including was well delineated. However, I have some comments:

Response:

 We would like to thank the reviewer for the kind words and support.

Comment (1): In my opinion, title is good, but it should be changed to be more attractive.

Response:

We agree with the reviewer and the title has been adjusted to be "Wasp venom biochemical components and their potential in biological applications and nanotechnological interventions"

Comment (2): General proofreading is highly recommended.

Response:

Authors read the manuscript thoroughly and amend it accordingly. 

Comment (3): Introduction. There is a brief review of existing knowledge and relevance of study.

Response:

We agree with the reviewer and hence addressed this issue,

“However, their peptides have been presented in trace quantities. Solid phase peptides synthesis (SPPS) was attributed to the design and development of these molecules [8]. Successfully, several peptides and their analogues were synthesized via SPPS technology such as mastoparan[9], anoplin [10],decoralin [11], polybia-MP-I [12], polybia-CP [13,14], polydim-I [15], and agelaia-MP [16]. The synthetic peptides have antimicrobal, and anticancer properties [17,18].

The nests and venoms of wasps have been their role in the synthesis of nanoparticles of gold and silver tested. These nanoparticles were proven effective as antimicrobial and anticancer entities against a variety of microorganisms and cancer cells [19–21].”

“VIT is the most effective method known so far for the avoidance of the systemic sting reactions even after discontinuation of the therapy [24].”

Comment (4): I recommend to the authors to merge the two sub-titles, 2. Biological properties of wasp venom and 3. Isolated and synthesized bioactive peptides from wasp venom, together because they give the same meaning and can be one section.

Response:

We would like to thank the referee once again for this comment; the two sections have been merged.

Comment (5): Figures and legends. Legends are adequate and figures are necessary to understand the results obtained. Authors need to make a space between the compounds in figure 1 and mark each one by A, B, C, …..

 Response:

This issue has been addressed and changes have been made.

Round 2

Reviewer 2 Report

The Authors addressed all my previous comments in this improved version. I recommend to accept this MS in the present form.